# Quantitative analysis of heparan sulfate using isotopically labeled calibrants

Zhangjie Wang[1], Katelyn Arnold [1], Yongmei Xu[1], Vijayakanth Pagadala [2], Guowei Su[1], Hannah Myatt[1], Robert J. Linhardt [3] & Jian Liu [1✉]

Heparan sulfate is a sulfated polysaccharide that displays essential physiological functions. Here, we report a LC-MS/MS-based method for quantitatively determining the individual disaccharide composition and total amount of heparan sulfate. Using eight $^{13}$C-labeled disaccharide calibrants and one $^{13}$C-labeled polysaccharide calibrant, we complete the analysis in one-pot process. The method is both sensitive and has the throughput to analyze heparan sulfate from mouse tissues and plasma.

---

[1] Division of Chemical Biology and Medicinal Chemistry, Eshelman School of Pharmacy, University of North Carolina, Chapel Hill, NC 27599-7355, USA. [2] Glycan Therapeutics, LLC, 617 Hutton Street, Suite 103, Raleigh, NC 27606, USA. [3] Department of Chemistry and Chemical Biology, Center for Biotechnology and Interdisciplinary Studies, Rensselaer Polytechnic Institute, Troy, NY 12180, USA. ✉email: jian_liu@unc.edu

Heparan sulfate is an essential glycan and present on the cell surface and in the extracellular matrix, participating in a wide range of biological functions[1]. It is a highly heterogeneous polysaccharide that contains a disaccharide repeating unit of glucuronic acid (GlcA) or iduronic acid (IdoA) and glucosamine (GlcN) residues, and each is capable of carrying sulfo groups. The sulfation patterns in heparan sulfate govern its binding to growth factors, protein inhibitors, and chemokines to regulate embryonic development[2], and to control blood coagulation[3] and inflammatory responses[4]. A reliable and sensitive method to determine the structure and the quantity of heparan sulfate under different physiological and pathophysiological conditions is highly desirable. Disaccharide compositional analysis is the most widely used technique for the conformation of heparan sulfate structure and total content determination as the analysis is capable of handling saccharide sequence heterogeneity[5,6]. This process relies on a complete depolymerization of heparan sulfate polysaccharides into disaccharides. By measuring the amount of the resultant disaccharide building blocks, one can obtain disaccharide composition for structural information and the total amount of heparan sulfate by summing up the quantity of each disaccharide. Liquid chromatography–coupled mass spectrometry (LC–MS) method and LC–tandem mass spectrometry (LC–MS/MS) have played increasingly significant roles in elucidating heparan sulfate disaccharide composition due to their high sensitivity, selectivity, and accuracy[7,8]. Isotope-labeled *Escherichia coli* K5 polysaccharide by inserting $^{13}C/^{15}N$ into disaccharide units has been developed for investigations of protein/heparan sulfate interactions[9] and quantification analysis of heparan sulfate disaccharides[10]. Isotope dilution mass spectrometry technique has been applied for the quantification analysis of heparan sulfate disaccharides from biological samples[11,12]. Nevertheless, it is often quite challenging to perform quantitative analysis using the LC–MS or LC–MS/MS method due to the lack of appropriate internal reference standards.

## Results

**Preparation of $^{13}C$-labeled calibrants.** Two types of $^{13}C$-labeled calibrant standards were prepared for this study: eight disaccharide calibrants and one polysaccharide-recovery calibrant. The disaccharide calibrants were prepared from hexasaccharides (compounds **1** and **2**) and heparan sulfate octasaccharides (compounds **3** and **4**) after their digestion with heparin lyases (Fig. 1a). The site-specifically labeled [$^{13}C$]oligosaccharides were synthesized using a chemoenzymatic method as previously reported[13]. The molecular mass of disaccharide calibrants is 6 Da higher than their counterparts in heparan sulfate from biological sources. Chemical purity of each of the disaccharide calibrants and their isotopic purity were determined to be >95% and >99%, respectively (Supplementary Figs. 1–8). The eight disaccharides prepared in the present study correspond to most of the disaccharides within heparan sulfate (Supplementary Table 1)[14]. The polysaccharide-recovery calibrant is *N*-sulfo heparosan containing a disaccharide repeating unit of -GlcA–GlcNS- (Fig. 1b). Within the recovery calibrant, the GlcA and GlcNS residues carry [$^{13}C$] carbons that were introduced metabolically by *E. coli* K5 strain (Fig. 1b). Recovery calibrant ΔIVS is the only product after heparin lyase digestion, and the molecular mass is 12 Da higher than the unlabeled counterpart in heparan sulfate (Supplementary Fig. 9).

**Determination of linear dynamic range using calibrants.** We evaluated the suitability of using isotopically labeled standards for quantitative analysis through the LC–MS/MS method. The entire protocol involves three steps: (1) heparan sulfate extraction step,

(2) digestion of heparan sulfate by heparin lyase I–III followed by disaccharide derivatization, and (3) LC–MS/MS analysis (Fig. 2a). Disaccharide calibrants are added at step 2 to serve as internal standards. A chemical group, known as AMAC (3-aminoacridin-9-(10H)-one), is then coupled to disaccharides to form disaccharide–AMAC conjugates[15]. The results from the dynamic range study revealed that the method displays excellent linearity from the concentration of 1–800 μg ml$^{-1}$ for the average of all eight disaccharides in the presence of 10 μg ml$^{-1}$ of disaccharide calibrants (Fig. 2c). The linear dynamic range for each individual disaccharide calibrant is comparable to the average for the eight disaccharides (Supplementary Figs. 10–17). The wide dynamic range essentially eliminates the need for generating a standard curve to perform the quantitation analysis without compromising the accuracy and reproducibility.

**Comparison of recovery yields after DEAE column.** We next introduced the isotopically labeled polysaccharide-recovery calibrant at the extraction step, or Step 1 in Fig. 2a, to calibrate the recovery yield for purifying heparan sulfate from biological sources. A DEAE-column purification is the key point during extraction process that may result in sample loss. We compared the recovery yield of our recovery calibrant with heparan sulfate and heparin after DEAE-column purification to simulate the extraction process. The recovery yield after DEAE-column purification for the recovery calibrant was 93.9%, comparable to that for heparan sulfate (96.8%) and for heparin (97.9%), despite heparin having a higher sulfation level than heparan sulfate. These data suggest that the recovery calibrant is co-purified with heparan sulfate and heparin during the column purification, despite the differences in sulfation levels (Fig. 2d). The yield of the recovery calibrant is quantified by measuring the amount of disaccharide-recovery calibrant ΔIVS with a molecular mass that is 12 Da higher than unlabeled ΔIVS and 6 Da higher than disaccharide calibrant ΔIVS, offering a unique molecular mass marker in disaccharide analysis (Fig. 2b).

**Quantification of heparan sulfate from biological samples.** Next, we employed our method to measure the amount and composition of heparan sulfate from mouse tissues. We chose to compare heparan sulfate from the mice suffering from acute liver injury after acetaminophen (APAP) overdose and healthy mice. APAP is a widely used pain medication; however, overdose of acetaminophen leads to severe liver injury. APAP overdose is the leading cause of drug-induced liver injury in the United States and Europe. A recent study suggests that heparan sulfate intimately participates in physiological and pathophysiological responses after acetaminophen-induced liver injury[16], prompting us to investigate the structural changes in heparan sulfate after liver injury. A cohort of mice were administered with a sublethal dose of APAP. The extent of liver injury in APAP-overdose mice was evidenced by high plasma concentrations of alanine aminotransferase (ALT), a biomarker for liver injury (Supplementary Tables 2 and 3). We observed that APAP overdose elevated the amount of total heparan sulfate in the liver to 294 ± 54 from 215 ± 30 ng mg$^{-1}$, a 37% increase ($p = 0.0027$) (Fig. 3a, Supplementary Table 4), compared with the control group. For individual disaccharides, APAP overdose significantly increased the amount of five different disaccharides, including ΔIS, ΔIIS, ΔIVS, ΔIIA, and ΔIVA. The ΔIVS and ΔIVA are disaccharides without *O*-sulfation, and ΔIS, ΔIIS, and ΔIIA disaccharides carry 6-*O*-sulfation (Fig. 3b). The structural analysis data suggest that there is upregulation in heparan sulfate biosynthesis after liver injury, but the changes are primarily reflected in 6-*O*-sulfation. In contrast, there was no significant difference in total heparan sulfate or

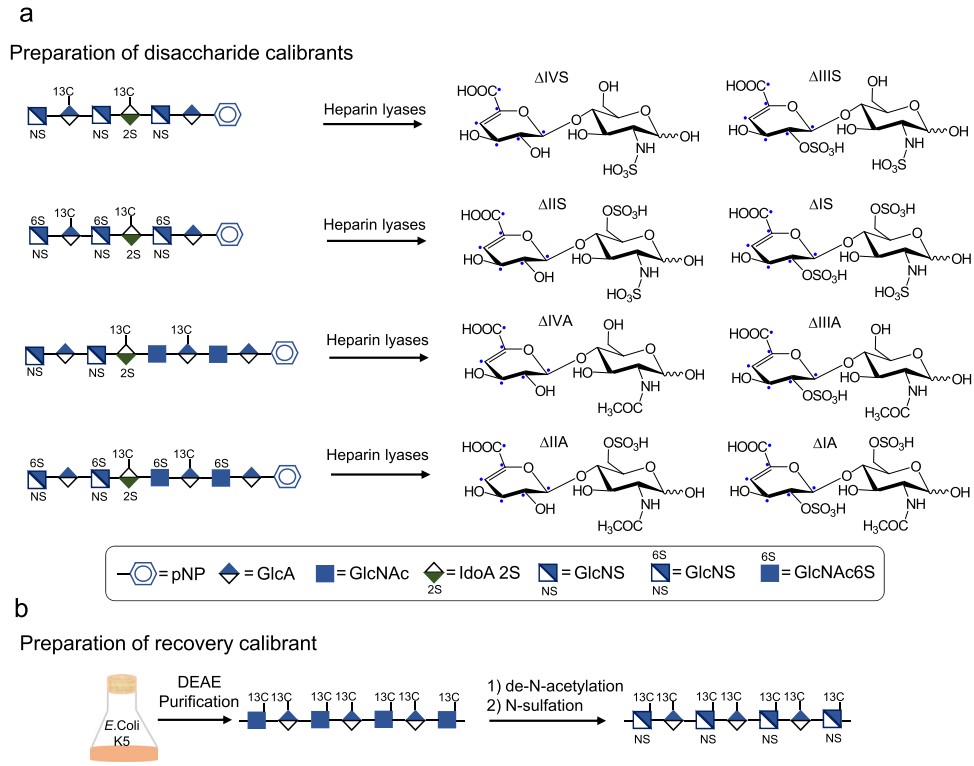

**Fig. 1 Schematic synthesis of disaccharide calibrants and one polysaccharide-recovery calibrant.** Panel **a** shows the procedures to prepare eight different disaccharide calibrants. Four $^{13}$C-labeled oligosaccharides were used to prepare individual disaccharides after the digestion with a mixture of heparin lyases, including heparin lyases I–III. Disaccharide calibrants were then purified by a Q-Sepharose column to obtain each individual disaccharide calibrant. Panel **b** shows the steps involving the synthesis of polysaccharide-recovery calibrant. *E. coli* K5 strain was grown in the M9 medium supplemented with [$^{13}$C] glucose at 37 °C overnight. $^{13}$C-labeled heparosan was purified from the culture media by a DEAE column. Heparosan was then subjected to de-*N*-acetylation reaction under a basic condition followed by *N*-sulfation using sulfur trioxide to yield the recovery calibrant.

individual disaccharide component in the heparan sulfate from lung and kidney (Supplementary Figs. S18–S19; Supplementary Tables 5 and 6).

**Quantification of hexasaccharide in the plasma.** Last, we used the method to determine the plasma concentration of an anticoagulant heparan sulfate hexasaccharide (6-mer) (Fig. 3c), which is a promising candidate for a new anticoagulant drug to replace animal-sourced heparin[17]. A method for measuring plasma concentration of 6-mer is critically important for the impending clinical evaluation. We used a $^{13}$C-labeled 6-mer as internal standard to calibrate the amount of unlabeled 6-mer. Because 6-mer contains a 3-*O*-sulfated residue that leads to resistance to the digestion[18], the digestion of 6-mer with heparin lyases yielded one disaccharide (ΔIS) and a trisaccharide (T3S) as expected (Fig. 3c). The ΔIS disaccharide and T3S trisaccharide were well resolved under the LC–MS/MS conditions (Fig. 3d). These two fragments offered us two internal standards to determine the amount of 6-mer. Indeed, the results show that using a fixed concentration of internal $^{13}$C-labeled 6-mer (4 µg mL$^{-1}$) can determine a range of concentrations of the unlabeled 6-mer (1–32 µg mL$^{-1}$), based on either ΔIS or T3S (Fig. 3e). Because T3S trisaccharide contains a 3-*O*-sulfated glucosamine residue that is a rare component in heparan sulfate from biological sources, our results suggest that the LC–MS/MS method is fully capable of analyzing 3-*O*-sulfation if the concentration is sufficiently high. It is important to note that direct quantitative measurement of intact 6-mer, without heparin lyase digestion and AMAC derivatization, reduced the detection sensitivity.

## Discussion

Here, we present a LC–MS/MS-based method to conduct quantitative analysis of heparan sulfate from biological sources. The crucial innovation in this method is to utilize a set of eight $^{13}$C-labeled disaccharide calibrants and one *N*-sulfo heparosan polysaccharide-recovery calibrant as internal standards. The combination of both disaccharide calibrants and recovery calibrant provides the sensitivity and throughput to study dynamic changes of heparan sulfate in a wide array of biological samples. One potential concern is over the accessibility to $^{13}$C-labeled carbohydrate standards. In our hand, the synthesis of disaccharide calibrants can be completed in 1.5–50 mg scale, and 1 mg of a disaccharide calibrant is adequate to carry out 4000 analyses. Large-scale synthesis of disaccharide calibrants can be achieved by further increasing the scale of $^{13}$C-oligosaccharide synthesis using our chemoenzymatic method. The synthesis of the polysaccharide-recovery calibrant is even easier than the synthesis for disaccharide calibrants because the recovery calibrant is made through bacterial fermentation. Using isotopically labeled proteins/peptides for quantitative proteomic analysis is becoming routine and has greatly accelerated biomedical research[19]. We anticipate that this new analytical tool will play a significant role toward understanding the relationship between structure and functions for heparan sulfate polysaccharides.

## Methods
**Chemoenzymatic synthesis of $^{13}$C-labeled oligosaccharides (compounds 1–4).** The structures of $^{13}$C-labeled oligosaccharides were appropriately designed, allowing us to isolate all eight disaccharide-calibrant targets carrying $^{13}$C-labeled $\Delta_{4,5}$-unsaturated uronic acid (ΔUA) after heparin lyase digestion. We intended to

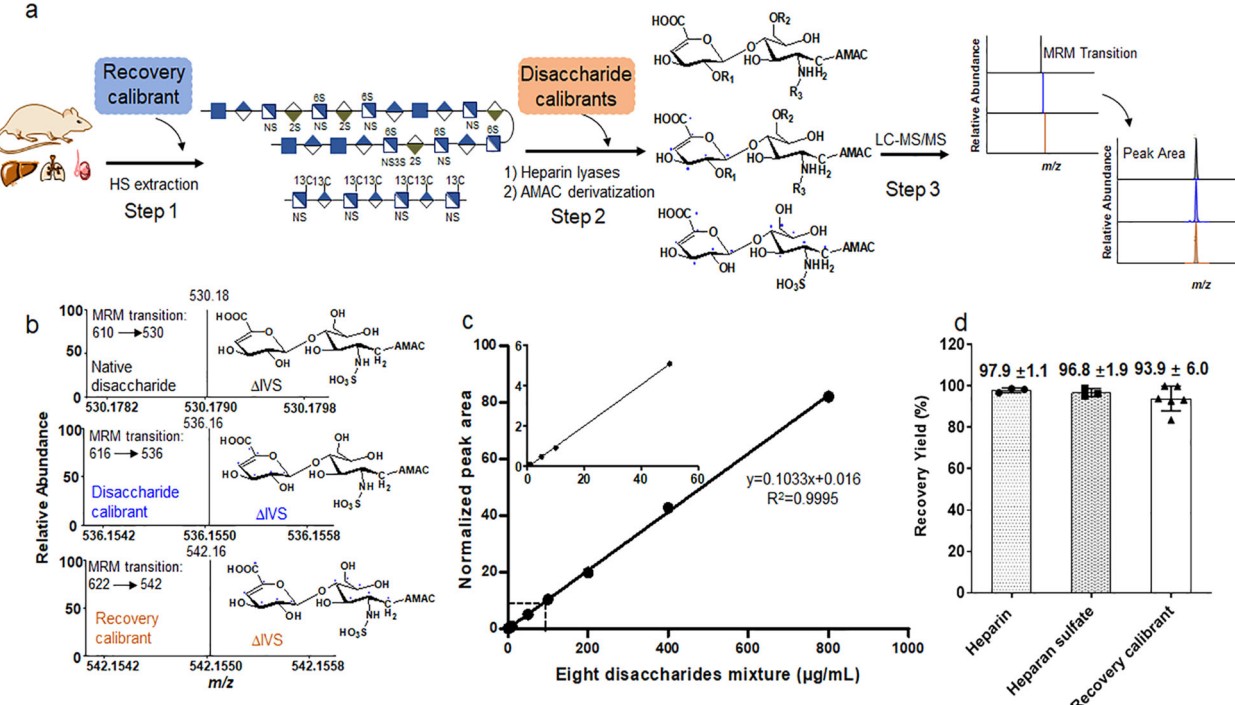

**Fig. 2 Validation of disaccharide calibrants and recovery calibrant in the protocol to analyze heparan sulfate.** Panel **a** shows the flowchart to conduct disaccharide compositional analysis. Three steps are involved. Recovery calibrant is added at the heparan sulfate extract stage, or Step 1. Disaccharide calibrants are added at the heparin lyase digestion stage, or Step 2. $R_1 = $ H or $SO_3H$, $R_2 = $ H or $SO_3H$, and $R_3 = COCH_3$ or $SO_3H$. Panel **b** shows MRM (multiple-reaction monitoring) ion chromatograms of ΔIVS disaccharide in three isotopically labeled formats: unlabeled, disaccharide calibrant ΔIVS (+6 Da), and recovery calibrant ΔIVS (+12 Da). Panel **c** shows the dynamic range of the analysis in the presence of disaccharide calibrants. Disaccharide calibrants (10 µg mL$^{-1}$) were mixed with various concentrations of unlabeled disaccharides followed by LC–MS/MS analysis. The peak area at each concentration was normalized to the area of disaccharide calibrants. The inset shows the response at the low concentration of unlabeled disaccharides. Data represent means ± S.D. Panel **d** shows the recovery yields heparin ($n = 3$), heparan sulfate (from bovine kidney) ($n = 3$), and the polysaccharide-recovery calibrant ($n = 6$) after DEAE-column purification. The error bars are shown as S.D.

minimize the number of [13]C-labeled oligosaccharides to prepare the disaccharides. In the meantime, we also chose the structures of oligosaccharides that would yield the disaccharides that can be readily separated by a Q-Sepharose column (GE Health) after heparin lyase digestion. To these ends, we synthesized four different oligosaccharides, and each oligosaccharide yielded two [13]C-labeled disaccharide calibrants.

The synthesis of compounds **1** and **3** started from GlcA-pNP. Five elongation steps, one detrifluoroacetylation/N-sulfation step, and one C5-epimerization/2-O-sulfation step were involved to synthesize compound **1**. The backbone GlcA*-GlcNTFA-GlcA*-GlcNTFA-GlcA-pNP (where GlcA* indicates universally [13]C-labeled GlcA residue) was synthesized in four elongation steps with heparosan synthase-2 (PmHS2) from *Pasteurella multocida*. In the elongation of GlcA-pNP to GlcNTFA-GlcA-pNP, the GlcA-pNP (50 mg) was incubated with PmHS2 (60 µg mL$^{-1}$) and 0.5 mM UDP-NTFA in a buffer containing 25 mM Tris (pH 7.5) and 15 mM MnCl$_2$ in a total volume of 100 mL. The reaction mixture was incubated at 37 °C overnight. The introduction of GlcA* residue to the disaccharide substrate (GlcNTFA-GlcA-pNP) was completed by incubating with PmHS2 (60 µg mL$^{-1}$) and 0.4 mM UDP-[13]C]GlcA in a buffer containing 25 mM Tris (pH 7.5) and 5 mM MnCl$_2$ in a total volume of 400 mL. The reaction mixture was incubated at 37 °C overnight. The polyamine II column (PAMN-HPLC (4.6 mm × 250 mm, YMC)) was applied to monitor the elongation-reaction degree. Once the reaction was complete, the reaction mixture was subjected to C18 column (0.75 × 20 cm, Biotage) purification. The molecular weight (MW) was characterized using ESI-MS. The pentasaccharide backbone was completed through addition of GlcNTFA and [13]C-labeled GlcA residues one additional time. The de-N-trifluoroacetylation of pentasaccharide backbone was implemented by suspending in 0.1 M LiOH and incubation on ice for 30 min. The completion of de-N-trifluoroacetylation was monitored by PAMN-HPLC column and ESI-MS. The N-sulfation of pentasaccharides was followed by incubating with NST (30 µg mL$^{-1}$), 50 mM MOPS (pH 7.0), and 0.5 mM PAPS at 37 °C overnight. The reaction completion was monitored using anion-exchange HPLC (TSKgel DNA-NPR-column (4.6 mm × 7.5 cm, 2.5 µm, Tosoh Bioscience)). The N-sulfated product was purified by Q-Sepharose (GE Health Care). Another GlcNTFA was introduced into the GlcA*-GlcNS-GlcA*-GlcNS-GlcA-pNP pentasaccharide using PmHS2 (60 µg mL$^{-1}$) following the procedure described above. The reaction mixture was purified by Q-Sepharose fast-flow column (GE Health Care), followed by N-detrifluoroacetylation/

N-sulfation described above. The product GlcNS-GlcA*-GlcNS-GlcA*-GlcNS-GlcA-pNP was further subjected to epimerization and 2-O-sulfation by incubating in 25 mM Tris (pH 7.5), C5-epimerase (3 µg mL$^{-1}$), 2-O-sulfotransferase (2-OST) (6.5 µg mL$^{-1}$), and 0.2 mM PAPS at 37 °C overnight. The reaction mixture was then purified by Q-Sepharose column to obtain compound **1**.

Seven elongation steps with UDP-GlcNAc, UDP-[13]C]GlcA, UDP-GlcNTFA, and UDP-GlcA, one detrifluoroacetylation/N-sulfation step, and one C5-epimerization/2-O-sulfation step were required to synthesize compound **3**. The synthesis of the backbone GlcNTFA-GlcA-GlcNTFA-GlcA*-GlcNAc-GlcA*-GlcNAc-GlcA-pNP was initiated from GlcA-pNP with PmHS2. The disaccharide GlcNAc-GlcA-pNP was synthesized by incubating the GlcA-pNP (50 mg), PmHS2 (60 µg mL$^{-1}$), and 0.5 mM UDP-NAc in a buffer containing 25 mM Tris (pH 7.5) and 15 mM MnCl$_2$ in a total volume of 80 mL. The reaction mixture was incubated at 37 °C overnight. The introduction of UDP-[13]C]GlcA to the disaccharide substrate (GlcNTFA-GlcA-pNP) was performed by incubating with PmHS2 (60 µg mL$^{-1}$) and 0.4 mM UDP-[13]C]GlcA in a buffer containing 25 mM Tris (pH 7.5) and 5 mM MnCl$_2$ in a total volume of 400 mL. The reaction mixture was incubated at 37 °C overnight. PAMN-HPLC was applied to monitor the extent of elongation-reaction degree and a C18 column (0.75 × 20 cm, Biotage) was applied for purification. The MW determination of the resulting trisaccharide intermediate was performed using ESI-MS. The synthesis of pentasaccharide backbone GlcA*-GlcNAc-GlcA*-GlcNAc-GlcA-pNP was completed by addition of GlcNAc and [13]C-labeled GlcA residues one more time. The GlcNTFA-GlcA-GlcNTFA-GlcA*-GlcNAc-GlcA*-GlcNAc-GlcA-pNP backbone was obtained by adding GlcNTFA and GlcA residues alternately with PmHS2 based on the protocol described above. The backbone was purified with a C18 column and dried, then dissolved in 0.1 M LiOH, and incubated on ice for 30 min to de-N-trifluoroacetylate. The de-N-trifluoroacetylation reaction was monitored by the PAMN-HPLC column and ESI-MS. The N-sulfation was accomplished by incubating with NST (30 µg mL$^{-1}$), 50 mM MOPS (pH 7.0), and 1 mM PAPS at 37 °C overnight. The reaction completion was monitored using TSKgel DNA-NPR column. N-sulfated product was purified by Q-Sepharose fast-flow column. The N-sulfation product was then subjected to epimerization and 2-O-sulfation by incubating in 25 mM Tris (pH 7.5), C5-epimerase (3 µg mL$^{-1}$), 2-OST (6.5 µg mL$^{-1}$), and 0.2 mM PAPS at 37 °C overnight. The reaction mixture was then purified using a Q-Sepharose fast-flow column to obtain compound **3**.

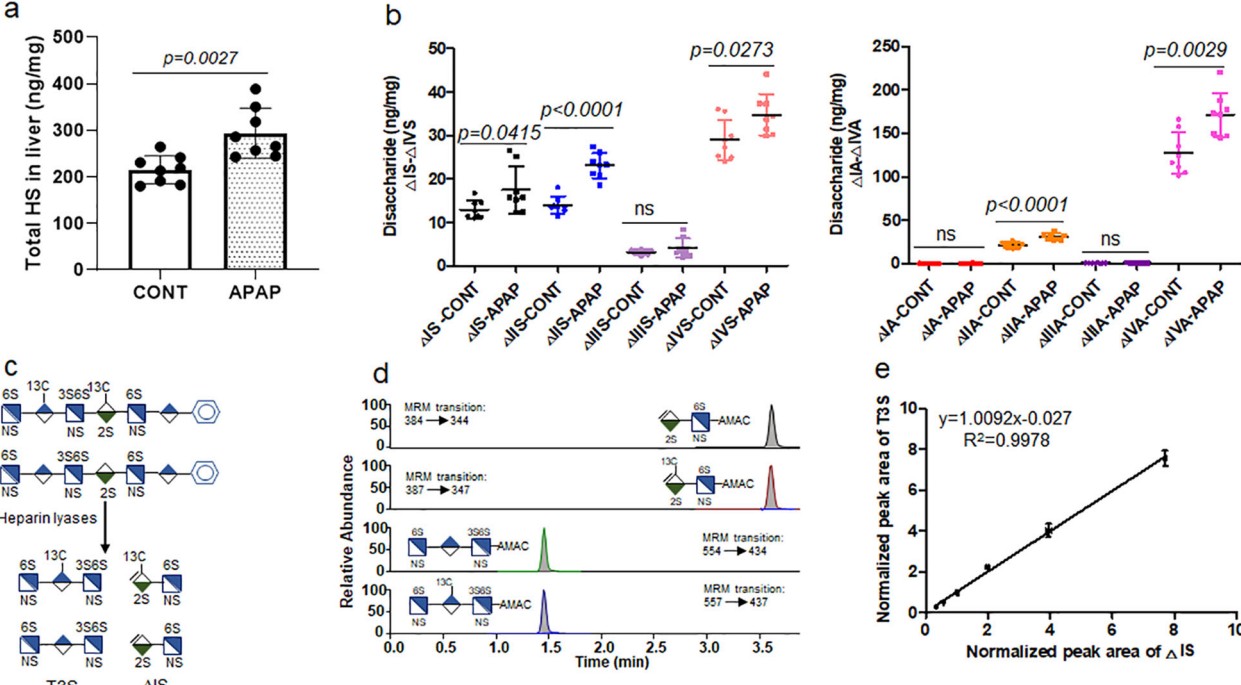

**Fig. 3 Demonstration of the utilities of quantitative LC/MS method in heparan sulfate and 6-mer under different biological contexts.** Panel **a** shows the total amount of heparan sulfate in the liver of mice with or without APAP overdose. The data are presented as mean ± S.D. ($n = 8$). Cont represents the group of animals without APAP overdose, and APAP represents the group of animals with APAP overdose. The $p$ value was determined by two-tailed unpaired $t$ test, ns, not significant ($p > 0.05$). Panel **b** shows the amount of individual disaccharides from mouse liver with or without APAP overdose. Panel **c** shows the procedures used to analyze 6-mer. A $^{13}$C-labeled 6-mer was used in the analysis. Panel **d** shows the MRM (multiple-reaction monitoring) ion chromatograms of ΔIS disaccharide (both unlabeled and $^{13}$C-labeled) and T3S trisaccharide (both unlabeled and $^{13}$C-labeled). Disaccharide and trisaccharide were well resolved by our LC method. Panel **e** shows the correlated linearity range for the analysis of 6-mer using two calibrants, the error bars are shown as S.D. The amounts of unlabeled 6-mer calibrated by ΔIS disaccharide or by T3S were nearly identical.

To synthesize compound **2**, compound **1** (30 mg) was incubated with 6-O-sulfotransferase 3 (6-OST-3) (900 µg mL$^{-1}$), 50 mM MOPS (pH 7.0), and 0.25 mM PAPS in a total volume of 500 mL at 37 °C overnight. The final product compound **2** was purified by Q-Sepharose fast-flow chromatography. To synthesize compound **4**, compound **3** (24 mg) was incubated with 6-OST-3 (900 µg mL$^{-1}$) and 0.3 mM PAPS in a buffer containing 50 mM MOPS (pH 7.0) in a total volume of 500 mL at 37 °C overnight. The reaction completion was monitored using TSKgel DNA-NPR column, and the product was purified by Q-Sepharose fast-flow column.

All compounds were purified by Q-Sepharose fast-flow chromatography. Mobile phase A was 20 mM NaOAc, pH 5.0, mobile phase B was 20 mM NaOAc, pH 5.0, and 1 M NaCl. The linear- gradient elution was used to wash the compounds from the column, and the elution gradient was changed according to the size and sulfo group numbers of compounds. All the compounds eluting from Q-Sepharose column were further dialyzed against deionized water using a 1000-molecular- weight cutoff (MWCO) membrane.

The synthesis of UDP-[$^{13}$C]GlcA was completed enzymatically from [$^{13}$C] glucose as described previously[13,20].

**Preparation of $^{13}$C-labeled N-sulfo heparosan from E. coli K5.** The expression of $^{13}$C-labeled *E. coli* K5 heparosan was carried out in the M9 media[21]. The stock 10× M9 salts were prepared by adding 67.6 g of Na$_2$HPO$_4$, 30.0 g of KH$_2$PO$_4$, and 5.0 g of NaCl in 1 L of water with pH 7.4. The *E. coli* K5 strain (from American Type Culture Collection) was grown in the 1 L M9 media containing 100 mL of 10× M9 salts, 1.0 g of NH$_4$Cl, 900 mL of H$_2$O, 10.0 mg of FeSO$_4$·7H$_2$O, 2.0 g of [$^{13}$C]glucose (Cambridge Isotope Laboratories), 2.0 mL of 1 M MgSO$_4$, 0.1 mL of 1 M CaCl$_2$, and 10.0 mg of thiamine hydrochloride at 37 °C on a shaker overnight. The medium was incubated in the shaker at 37 °C overnight. The supernatant was harvested by centrifugation at 7000 rpm for 30 min, then filtered through a 0.22-µm membrane. The pH of the filtrate was adjusted to 4.0 with acetic acid before further purification. The DEAE column was used to purify the K5 polysaccharide, heparosan, with buffer A containing 20 mM NaOAc, 50 mM NaCl, pH 4.0, and buffer B containing 20 mM NaOAc, 1 M NaCl, pH 4.0. After loading the medium, 1 L buffer A was applied to wash the column at a flow rate of 4 mL min$^{-1}$. Buffer B was then used to elute heparosan. The eluent was mixed with EtOH (1:2, v/v) to precipitate heparosan at 4 °C overnight in an explosion-proof refrigerator. The mixture was spun down at 7000 rpm for 30 min, and the precipitate was resuspended in water. Heparin lyase III was applied to digest heparosan, and the disaccharide composition of the resulting

mixture was characterized by LC–MS. *N*-sulfation of heparosan required two chemical reactions, chemical de-*N*-acetylation and *N*-sulfation. The heparosan was dissolved in 2 M NaOH and incubated at 55 °C overnight. After incubation, the solution was neutralized to pH 7.0 with HCl and dialyzed against water with a 1000- MWCO membrane. The dialyzed sample was dried and resuspended in 3 mL of H$_2$O, followed by adding 5.0 mg of Na$_2$CO$_3$, and 5 mg of sulfur trioxide:triethylamine complex was added. The pH of solution was maintained at 9.5 with acetic acid and NaOH. The solution was then incubated at 48 °C overnight. After incubation, the reaction mixture was neutralized using acetic acid to pH 7.0, following which the DEAE-column purification was performed as described above. The recovered product was dialyzed against water with a 1000-MWCO membrane. The structural characterization of *N*-sulfo heparosan was accomplished using heparin lyase I, II, and III digestion followed by LC–MS analysis.

**Preparation of $^{13}$C-labeled heparan sulfate disaccharide calibrants.** Heparin lyases I, II and III expressed from *Flavobacterium heparinum* expressed in *E. coli* were used to cleave compounds **1–4** to prepare the eight $^{13}$C-labeled disaccharide calibrants. The heparin lyase enzymatic solution contained 50 µL of substrate (compounds **1–4**), 175 µL of enzymatic buffer (100 mM sodium acetate/2 mM calcium acetate buffer (pH 7.0) containing 0.1 g L$^{-1}$ bovine serum albumin (BSA)), and 48 µL of an enzyme cocktail containing 5 mg ml$^{-1}$ each of heparin lyases I–III. The reaction mixture was incubated at 37 °C overnight. The extent of reaction completion was monitored by the strong anion-exchange chromatography on a Pro Pac PA1 column (9 × 250 mm, Thermo Fisher Scientific) by measuring the absorbance at 232 nm. The purification of $^{13}$C-labeled disaccharide calibrants was performed on a Q-Sepharose fast-flow column. Mobile phase A was 20 mM NaOAc, pH 5.0, and mobile phase B was 20 mM NaOAc and 1 M NaCl, pH 5.0. The elution gradient based on the number of sulfate groups of disaccharide calibrants with a flow rate of 1 mL min$^{-1}$ was used. The absorbance at 232 nm was scanned and recorded. After purification, the disaccharides were desalted on a Sephadex G-10 column. The quantification of $^{13}$C-labeled disaccharide calibrants was performed based on the standard curve of commercially available native heparan sulfate disaccharide standards (Iduron).

**Structure analysis of $^{13}$C-labeled disaccharide calibrants.** A strong anion-exchange column Pro Pac PA1 (9 × 250 mm, Thermo Fisher Scientific) was used to determine the purity of $^{13}$C-labeled disaccharides after purification. Mobile phase A

was 3 mM $NaH_2PO_4$, pH 3.0. Mobile phase B was 3 mM $NaH_2PO_4$ and 1 M NaCl, pH 3.0. The gradient was as follows: 0–20 min 5–20% B, 20–72 min 20–95% B, and 72–75 min 95–100% B with a flow rate of 1 mL min$^{-1}$. The ultraviolet absorbance at 232 nm was scanned and recorded. The different retention times of two groups of isomers (Δ[$^{13}$C]UA2S-GlcNS and Δ[$^{13}$C]UA-GlcNS6S; Δ[$^{13}$C]UA2S-GlcNAc and Δ[$^{13}$C]UA-GlcNAc6S) on SAX-HPLC were determined by comparing with the retention times of the native disaccharide standards (Iduron) on SAX column.

ESI-MS (Thermo Scientific TSQ Fortis) analysis was used to confirm the MW of each $^{13}$C-labeled disaccharide. The ESI-MS analysis was performed in the negative-ion mode and with the following parameters: Neg ion spray voltage at 3.0 kV, sheath gas at 15 Arb, ion transfer tube temp at 320 °C, and vaporizer temp at 100 °C. The mass range was set at 240–800.

**Linear dynamic range determination**. Individual stock solutions of eight heparan sulfate-native disaccharides (Iduron) were prepared in water at 1 mg mL$^{-1}$. A stock solution of the mixture of the eight native disaccharides, each at 1 mg mL$^{-1}$, was obtained by mixing an equal volume of eight individual stock solutions. The linear dynamic range of the working solutions was determined by a serial dilution of the mixture stock solution in water to obtain the final concentrations of 1, 5, 10, 50, 100, 200, 400, and 800 μg mL$^{-1}$. The mixture stock solution of eight $^{13}$C-labeled disaccharide calibrants at concentration of 1 mg mL$^{-1}$ was diluted to 10 μg mL$^{-1}$ and added to the linear dynamic range working solutions as an internal standard. The linear dynamic range working solutions containing $^{13}$C-labeled internal standard were freeze-dried and reconstituted in the 20 μL of mouse plasma. The reconstituted solutions were filtered by passing through a YM-3 kDa spin column (Millipore) and washed twice with deionized water to recover the disaccharides in the eluent. The recovered disaccharides were lyophilized. The AMAC derivatization of lyophilized disaccharides was carried out by adding 10 μL of 0.1 M AMAC solution in DMSO/glacial acetic acid (17:3, v/v) and incubating at room temperature for 15 min. Then 10 μL of 1 M aqueous sodium cyanoborohydride (freshly prepared) was added to this solution. The reaction mixture was incubated at 45 °C for 4 h. After incubation, the reaction solution was centrifuged to obtain the supernatant that was subjected to the LC–MS/MS analysis. Three replicates of each concentration were performed on LC-MS/MS analysis. After LC–MS/MS analysis, the peak area of native/unlabeled disaccharide was normalized to the peak area of the corresponding $^{13}$C-labeled disaccharide. The normalized peak area was plotted against the anticipated concentrations of native/unlabeled disaccharide. The error bars represent the standard deviation (S.D.) of the data.

**LC-MS/MS analysis**. The analysis of AMAC-labeled disaccharides was performed on a Vanquish Flex UHPLC System (Thermo Fisher Scientific) coupled with TSQ Fortis triple-quadruple mass spectrometry as the detector. The C18 column (Agilent InfinityLab Poroshell 120 EC-C18 2.7 μm, 4.6 × 50 mm) was used to separate the AMAC-labeled disaccharides. Mobile phase A was 50 mM ammonium acetate in water. Mobile phase B is methanol. The elution gradient from 5 to 45% mobile phase B in 10 min, followed by isocratic 100% mobile phase B in 4 min and then isocratic 5% mobile phase B in 6 min, was performed at a flow rate of 0.3 ml/min. Online triple–quadruple mass spectrometry operating in the multiple-reaction-monitoring (MRM) mode was used as the detector. The ESI–MS analysis was operated in the negative-ion mode using the following parameters: Neg ion spray voltage at 4.0 kV, sheath gas at 45 Arb, aux gas 15 arb, ion transfer tube temp at 320 °C, and vaporizer temp at 350 °C. TraceFinder software was applied for data processing.

**Mouse model of APAP liver injury**. All animal experiments were approved by the Institutional Animal Care and Use Committee (IACUC) of the University of North Carolina at Chapel Hill. C57BL/6J mice were fasted overnight (12–15 h) to deplete glutathione stores before acetaminophen (APAP) (Sigma) administration. Fresh APAP was dissolved in warm (~50 °C) sterile saline, cooled to 37 °C, and injected intraperitoneally at 400 mg/kg. As a positive control, mice were injected with saline intraperitoneally. Plasma collected 24 h after injection was measured for ALT concentration using the ALT Infinity reagent (Thermo Fisher) following the manufacturer's instructions.

**Extraction and quantitation analysis of heparan sulfate from the murine tissues**. Tissue organs, including the liver, kidney, and lung, were harvested from the saline control mice and APAP-overdose mice, respectively. Liver heparan sulfate was extracted from eight saline control mice and eight APAP-overdose mice liver tissues. Kidney and lung heparan sulfate was isolated from five saline control mice and five APAP-injured mice, respectively. Heparan sulfate was purified from the whole livers, left kidneys, and right lungs, respectively. Tissues were excised, homogenized, and defatted by suspension and vortex in chloroform/methanol mixtures (2:1, 1:1, and 1:2 (v/v)). The dried and defatted tissues were digested with Pronase E (10 mg:1 g (w/w), Pronase E/tissue) at 55 °C for 24 h to degrade the proteins. Two microliters of $^{13}$C-labeled polysaccharide-recovery calibrant was added into the digestion solution. Heparan sulfate was recovered from the digested solution using a DEAE column. DEAE column mobile phase A was 20 mM Tris, pH 7.5 and 50 mM NaCl, and mobile phase B was 20 mM Tris, pH 7.5 and 1 M NaCl. After loading the digested solution, the column was washed with 10-column

volumes of buffer A to discard the contaminants, followed by 10-column volumes of buffer B to elute the heparan sulfate fraction. The heparan sulfate eluting from the DEAE column was desalted using a YM-3 kDa spin column and washed three times with deionized water to remove salt. The 200 μL of enzymatic buffer (100 mM sodium acetate/2 mM calcium acetate buffer (pH 7.0) containing 0.1 g L$^{-1}$ BSA), and 60 μL of enzyme cocktails containing 5 mg/ml each of heparin lyase I, II, and III, were added to digest the retentate on the filter unit of the YM-3 kDa column. The column was incubated at 37 °C overnight. Before recovering the disaccharides from the digest solution, a known amount of $^{13}$C-labeled disaccharide calibrants was added to the digestion solution. The heparan sulfate disaccharides and disaccharide calibrants were recovered by centrifugation, and the filter unit was washed twice with 200 μL of deionized water. The collected filtrates were freeze-dried before the AMAC derivatization. The AMAC label and LC–MS/MS analysis of the collected disaccharides of tissues was performed as described above. The amount of tissue heparan sulfate was determined by comparing the peak area of native disaccharide to each disaccharide calibrant, and the recovery yield was calculated based on a comparison of the amount of recovery-calibrant disaccharide in the tissue samples and control, respectively.

**Determination of heparin and heparan sulfate recovery yield from DEAE column**. $^{13}$C-labeled polysaccharide-recovery calibrant as a control was applied to evaluate the heparin and heparan sulfate recovery yield from DEAE column. The recovery calibrant was spiked in the heparin or heparan sulfate solution with a certain volume, respectively. The mixture was divided into two aliquots, one for DEAE-column purification and another as control without DEAE-column purification. An aliquot of the mixture was purified with a DEAE–chromatography and YM-3 kDa desalting as described above, whereas the control was directly spiked in the buffer B, followed by its desalting by YM-3 kDa. The 200-μL enzymatic buffer (100 mM sodium acetate, 2 mM calcium acetate (pH 7.0) containing 0.1 g L$^{-1}$ BSA), and 30 μL of enzyme cocktails containing 5 mg/ml each of heparin lyase I–III, were added to digest the retentate on the filter unit of the YM-3 kDa column. The column was incubated at 37 °C overnight. Before recovering the disaccharides from the digest solution, a known amount of $^{13}$C-labeled disaccharide calibrants was added to the digestion solution. The heparin/heparan sulfate disaccharides and disaccharide calibrants were recovered by centrifugation, and the filter unit was washed twice with 200 μL of deionized water. The collected disaccharides were freeze-dried before the AMAC derivatization. The AMAC labeling and LC–MS/MS analysis of the collected disaccharides was performed as described above. The recovery calibrant and heparin/heparan sulfate recovery efficiency from DEAE column was determined by comparing the amount of recovery calibrant and heparin/heparan sulfate from DEAE column with that of the control.

**Quantification analysis of oligosaccharides in mouse plasma**. $^{13}$C-labeled heparan sulfate hexasaccharide (6-mer) was synthesized by inserting a 3-O-sulfo group with 3-OST-1 to compound **2**. Compound **2** (10 mg) was incubated with 1 mM PAPS in a solution containing 50 mM MOPS (pH 7.0), 10 mM $MnCl_2$, 5 mM $MgCl_2$, and 2 mL of 3-OST-1 (20 μg mL$^{-1}$) in a total volume of 100 mL. The reaction mixture was incubated at 37 °C overnight, following purification by a Q-Sepharose fast-flow column. The unlabeled 6-mer was diluted into a series of concentrations from 1 to 32 μg mL$^{-1}$; a 20-μL aliquot at each concentration was removed and mixed with 20 μL of $^{13}$C-labeled 6-mer with concentration of 4 μg mL$^{-1}$. The mixture was dried, and then 20 μL of mouse plasma was spiked into each tube to redissolve the oligosaccharides. The 180-μL enzymatic buffer (100 mM sodium acetate/2 mM calcium acetate buffer (pH 7.0) containing 0.1 g L$^{-1}$ BSA), and 60 μL of enzyme cocktail containing 5 mg/ml each of heparin lyase I–III, was added to digest the 6-mer at 37 °C overnight. After digestion, YM-3 kDa spin columns were used to recover the digests, and the columns were washed twice with 200 μL of deionized water. The filtrates were collected and freeze-dried. The AMAC labeling and LC–MS/MS analysis of the digests of 6-mer were carried out as described above. The normalized peak areas of T3S were plotted against the normalized peak areas of ΔIS.

**Statistics and reproducibility**. Quantification of heparan sulfate in the liver was performed by collecting tissues from eight APAP-overdose mice ($n = 8$) and eight healthy mice ($n = 8$), respectively. Determination of the amount of heparan sulfate from kidney and lung tissues was performed by collecting tissues from five APAP-overdose mice ($n = 5$) and five healthy mice ($n = 5$), respectively. Three replicates of each sample were performed on LC–MS/MS analysis. The data were presented as the mean ± S.D. The $p$ value was determined by two-tailed unpaired $t$ test. The statistical significance was defined as $p < 0.05$.

**Reporting summary**. Further information on research design is available in the Nature Research Reporting Summary linked to this article.

## Data availability

All the mass spectral data related to this research were deposited in the MassIVE [https://doi.org/10.25345/C5F419]. Source data are available as Supplementary Data. The relevant data would be available from the corresponding author upon reasonable request.

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

## Acknowledgements

This work is supported in part by NIH grants (HL094463, HL144970, GM128484, GM134738, and HL139187).

## Author contributions

Z.J.W. synthesized disaccharide calibrants and the polysaccharide-recovery calibrant. Z.J.W. also designed the project, completed the LC–MS/MS analysis, and wrote the paper. K.A. and H.M. prepared the heparan sulfate from healthy and APAP-overdose mice. Y.X. participated in the synthesis of heparan sulfate oligosaccharides and secured the funding to acquire the LC–MS/MS instrument. V.P. provided UDP-[13C]GlcA to synthesize heparan sulfate oligosaccharides. G.S. involved in structural analysis of oligosaccharides and disaccharide calibrants. R.J.L. and J.L. designed the project, interpreted the data, and wrote the paper. All authors participated in writing the paper.

## Competing interests

J.L. and Y.X. are founders for the Glycan Therapeutics, L.L.C. V.P. is an employee of the Glycan Therapeutics and has an equity. The remaining authors declare no competing interests.
