## [Peer Review File · Communications Biology]

Reviewers' comments:

Reviewer #1 (Remarks to the Author):

The manuscript described an LC-MS/MS method for the quantitation of heparan sulfate using isotopically labeled calibrants. The research is novel, logical, and authors conducted substantial amount of work and generate enough data in support of the conclusion. The reviewer has the following comments.

1. Panel D of the manuscript shows the recovery yields for HS, where bovine kidney HS was used for recovery evaluation. Did the authors tested one lot or different lots? Does the composition remain consistent if different lots of HS was tested?

2. This authors described the procedures for calibration standards preparations at page 8 of supporting information. When quantitating HS from unknown study samples, how was the samples processed? I assume only the internal standard is added? Please elaborate on this.

3. Was quality control samples introduced as part of the analysis? If the method is intended for research purpose only, the QC samples might not be used. However, if the method is used for diagnostic purposes or in support of pre-clinical and clinical studies under regulated settings in the future, QC samples, as a part of method full validation, should be mandatorily conducted.

4. Reference 5 is not sufficient, please include the following reference at a minimum:
a).<https://doi.org/10.1016/j.cca.2018.11.001> b). Biomed Chromatogr. 2018 Oct;32(10):e4294 c). Bioanalysis. 2016 Feb;8(4):285-95. doi: 10.4155/bio. etc

5. Figure 1A, duplicate GlcA was presented at the bottom of the figure

Reviewer #2 (Remarks to the Author):

The authors address the important need for absolute quantification methods to characterize heparin and heparan sulfate (HS). While several LC-MS methods exist, none have incorporated stable isotope labeled disaccharide standards. The use of such labeled disaccharides will significantly improve the ability to determine absolute quantities of disaccharides from biological samples. This solves a serious problem for understanding the roles of HS in biological processes. The choice to use reductively aminated disaccharides makes sense because of the stability and resolution of reversed phase liquid chromatography. The authors demonstrate the applicability of the method in a mouse liver injury model. They also show quantification of an anticoagulant HS hexamer in blood plasma. Such quantification is important for development of HS-based therapeutics but very challenging analytically. The use of stable isotope labeled hexamer is a rigorous and effective means of absolute quantification from complex biological backgrounds. The method development is rigorous and well described. In summary, this manuscript describes an important advance in quantitative characterization of HS from biological samples.

All mass spectral data should be posted to a public repository prior to publication.

P. 3. "However, it is often quite challenging to perform quantitative analysis using the LC-MS or LC-MS/MS method due to the lack of appropriate internal reference standards." Add a brief summary of the LC-MS methods for analysis HS disaccharides from biological tissue that are used most frequently in the current literature.

Reviewer #3 (Remarks to the Author):

Review for "Developing a quantitative method to analyze heparan sulfate using isotopically labeled calibrants"

This manuscript by Liu, Linhardt, and coworkers describes a strategy for quantitatively performing composition analysis of heparan sulfate. The authors report the preparation of ¹³C labeled disaccharides and polysaccharides that mimic the glycosaminoglycan chemical structure. They then show that these isotopically labeled small molecules can be used as internal standards for mass spectrometry analysis of heparan sulfate deriving from biological contexts. These demonstrations are unfortunately undermined by the manuscript's current framing that relegates key data figures to the supporting information and discusses some results only superficially. Furthermore, the manuscript lacks key citations that would be useful for identifying the novelty in the present work; perhaps most notably, an uncited 2009 report from this same collaboration that claims a similar advance. Elaboration of these comments and other points are delineated below to improve the quality of the manuscript:

- The manuscript is seemingly under-referenced with several other key reports of isotopically labeled heparan sulfate for structural biology and mass spectrometry analysis including: C. Laguri et al. *J. Am. Chem. Soc.* 2011, 133, 25, 9642-9645; H. Naimy et al. *Bioanalysis* 2016, 8, 4, 285-295; Z. Zhang et al. *Anal. Chem.* 2009, 81, 11, 4349-4355.
- SI Figure S18 should be moved to the main text to better highlight how the described method was applied to heparan sulfate from biological contexts (both mouse and plasma). These demonstrations are presumably one of the key advances of the present study, which should be reflected in the figure set.
- Some of the p values in these data figures should be checked. In particular, in SI Figure S18 the data for saccharide IVA appears to have a large enough standard deviation that the p value should be > 0.0029.
- Similarly, for SI Figure S19 "ns" should be clearly defined as a p value greater than a certain number, or the actual p values should just be provided for clarity.
- The current description of results from the plasma study unsatisfactorily leaves several questions unanswered: what was the "range of concentrations" one could determine of the heparan sulfate hexasaccharide? What constitutes a "sufficiently high" concentration?
- In the linear dynamic range curves, which establish a linear dependence of peak area on disaccharide concentration, why do some of the data points have error bars and others do not? Were only certain concentrations tested multiple times? While the linear fits are convincing the specifics of data acquisition should be more clearly discussed.
- The seven isotopically labeled sulfo-disaccharides exhibit highly variable retention times, ranging from 17-47 mins despite similar elution conditions—this should be explained since their isolation and purity is a key requirement for the manuscript.
- Several typographical errors were found, which should be corrected in the final version of the manuscript:
 - o Page 4: "most of the disaccharides" insert "the"
 - o Page 6: "leads to resistance to" change "resistant" to "resistance"
 - o Page 7: "Large-scale synthesis of DC..." delete "For" at the start of sentence
 - o Page 8: "role towards understanding the relationship" move "the"

Thank you for reviewing our manuscript submitted to *Communications Biology* for publication (COMMSBIO-20-1000-T). Below are our point-by-point responses to your questions.

Reviewer #1

1. “The manuscript described an LC-MS/MS method for the quantitation of heparan sulfate using isotopically labeled calibrants. The research is novel, logical, and authors conducted substantial amount of work and generate enough data in support of the conclusion. The reviewer has the following comments.”

We thank for Reviewer #1’s comments on our work

2. “Panel D of the manuscript shows the recovery yields for HS, where bovine kidney HS was used for recovery evaluation. Did the authors tested one lot or different lots? Does the composition remain consistent if different lots of HS was tested?”

The bovine kidney HS applied for the recovery evaluation was made in our lab ¹ and heparin was a commercial product (Sigma). Each sample was from a single batch. The quantification results of disaccharides from bovine kidney HS and heparin as follows (values expressed in mole %.), showing the difference in sulfation levels for HS and heparin:

Disaccharide composition of bovine kidney HS							
Δ IS	Δ IIS	Δ IIIS	Δ IVS	Δ IA	Δ IIA	Δ IIIA	Δ IVA
5.54±0.21	10.82±0.21	1.94±0.09	10.98±0.75	n.d.	11.90±0.45	n.d.	58.62±0.48
Disaccharide composition of Heparin							
Δ IS	Δ IIS	Δ IIIS	Δ IVS	Δ IA	Δ IIA	Δ IIIA	Δ IVA
60.94±0.52	15.57±0.28	3.95±0.06	3.68±0.01	1.35±0.09	4.07±0.21	1.76±0.36	8.2±0.57

The purpose of our experiment was to determine if recovery calibrant has similar recovery yield as heparin and heparan sulfate after DEAE purification. We found that the recovery yield after DEAE column purification for recovery calibrant was 93.9%, comparable to that for HS (96.8%) and for heparin (97.9%), despite that heparin contains a higher sulfation degree than the HS. Although different batches of HS isolated from bovine kidney may have different disaccharide compositions, we think such difference should not affect the recovery yield from DEAE column purification.

3. “This authors described the procedures for calibration standards preparations at page 8 of supporting information. When quantitating HS from unknown study samples, how was the samples processed? I assume only the internal standard is added? Please elaborate on this.

Our method involves use of two calibrants. Recovery calibrant (a ^{13}C -labeled structurally heterogeneous polysaccharide) was used to control the recovery yield at the isolation step (Step 1 described in Fig 1a). Disaccharide calibrants (^{13}C -labeled disaccharide standards) were used to control the efficiency at AMAC derivatization (Step 2 described in Fig 1a) and ionization step (Step 3 described in Fig 1a).

To quantify the recovery yield at the extraction step, ^{13}C -labeled recovery calibrant was added to the sample containing HS. The sample was then loaded onto the DEAE column to determine the recovery yield of HS from column elution. The known amount ^{13}C -labeled eight HS disaccharides applied as DCs were added into the digestion solution of HS by heparin lyases I, II and III before recovering the disaccharides from the digest solution using YM-3KDa column.

To perform the linear dynamic range determination, a series of known concentrations of native/unlabeled HS disaccharide standards (from Iduron) were mixed with a single concentration of ^{13}C -labeled disaccharide calibrants (DCs). After LC-MS/MS analysis, the peak area of native/unlabeled disaccharide was normalized to the peak area of the corresponding ^{13}C -labeled disaccharide. The normalized peak area was plotted against the anticipated concentrations of native/unlabeled disaccharide.

These statements are included under “Supplementary Information” in the revised manuscript.

4. “Was quality control samples introduced as part of the analysis? If the method is intended for research purpose only, the QC samples might not be used. However, if the method is used for diagnostic purposes or in support of pre-clinical and clinical studies under regulated settings in the future, QC samples, as a part of method full validation, should be mandatorily conducted.”

We agree with the reviewer’s suggestion for the quality control (QC) samples introduction for diagnostic purposes, preclinical and clinical studies. QC samples can be analyzed at regular intervals to evaluate the assay precision and bias. In our paper, we didn’t perform the QC samples due to this method is only applied for lab research.

5. “Reference 5 is not sufficient, please include the following reference at a minimum: a).<https://doi.org/10.1016/j.cca.2018.11.001> b). Biomed Chromatogr. 2018 Oct;32(10):e4294 c). Bioanalysis. 2016 Feb;8(4):285-95. doi: 10.4155/bio. etc.”

Those papers have been cited in the revised manuscript.

6. “Figure 1A, duplicate GlcA was presented at the bottom of the figure.”

The second GlcA has been deleted in the Figure 1A

Reviewer #2

7. “The authors address the important need for absolute quantification methods to characterize heparin and heparan sulfate (HS). While several LC-MS methods exist, none have incorporated stable isotope labeled disaccharide standards. The use of such labeled disaccharides will significantly improve the ability to determine absolute quantities of disaccharides from biological samples. This solves a serious problem for understanding the roles of HS in biological processes. The choice to use reductively aminated disaccharides makes sense because of the stability and resolution of reversed phase liquid chromatography. The authors demonstrate the applicability of the method in a mouse liver injury model. They also show quantification of an anticoagulant HS hexamer in blood plasma. Such quantification is important for development of HS-based therapeutics but very challenging analytically. The use of stable isotope labeled hexamer is a rigorous and effective means of absolute quantification from complex biological backgrounds. The method development is rigorous and well described. In summary, this manuscript describes an important advance in quantitative characterization of HS from biological samples.”

We appreciate the Reviewer #2’s comments on the novelty and importance of our work.

8. “All mass spectral data should be posted to a public repository prior to publication.”

We have uploaded all the mass spectral data related to this research in a public repository MassIVE [doi:10.25345/C5F419].

9. “P. 3. “However, it is often quite challenging to perform quantitative analysis using the LC-MS or LC-MS/MS method due to the lack of appropriate internal reference standards.” Add a brief summary of the LC-MS methods for analysis HS disaccharides from biological tissue that are used most frequently in the current literature.”

We agree with this suggestion, and we provided more detailed information about widely used LC-MS techniques for HS disaccharides analysis from biological tissue in the revised manuscript as follows.

“Liquid chromatography coupled mass spectrometry (LC-MS) method and LC tandem mass spectrometry (LC-MS/MS) have played increasingly significant roles in elucidating HS disaccharide composition due to their high sensitivity, selectivity and accuracy^{7,8}. Isotope labeled *Escherichia coli* K5 polysaccharide by inserting ¹³C/¹⁵N into disaccharide unite has been developed for investigations of protein/HS

interactions⁹ and quantification analysis of HS disaccharides¹⁰. Isotope dilution mass spectrometry technique has been applied for the quantification analysis of HS disaccharides from biological samples^{11,12}. Nevertheless, it is often quite challenging to perform quantitative analysis using the LC-MS or LC-MS/MS method due to the lack of appropriate internal reference standards.”

Reviewer #3

10. “This manuscript by Liu, Linhardt, and coworkers describes a strategy for quantitatively performing composition analysis of heparan sulfate. The authors report the preparation of ¹³C labeled disaccharides and polysaccharides that mimic the glycosaminoglycan chemical structure. They then show that these isotopically labeled small molecules can be used as internal standards for mass spectrometry analysis of heparan sulfate deriving from biological contexts. These demonstrations are unfortunately undermined by the manuscript’s current framing that relegates key data figures to the supporting information and discusses some results only superficially. Furthermore, the manuscript lacks key citations that would be useful for identifying the novelty in the present work; perhaps most notably, an uncited 2009 report from this same collaboration that claims a similar advance.”

We thank for the Reviewer #3’s suggestions, we have followed those suggestions in the revised manuscript.

11. “The manuscript is seemingly under-referenced with several other key reports of isotopically labeled heparan sulfate for structural biology and mass spectrometry analysis including: C. Laguri et al. *J. Am. Chem. Soc.* 2011, 133, 25, 9642-9645; H. Naimy et al. *Bioanalysis* 2016, 8, 4, 285-295; Z. Zhang et al. *Anal. Chem.* 2009, 81, 11, 4349-4355.”

We have cited those research papers in the revised manuscript.

12. “SI Figure S18 should be moved to the main text to better highlight how the described method was applied to heparan sulfate from biological contexts (both mouse and plasma). These demonstrations are presumably one of the key advances of the present study, which should be reflected in the figure set.”

We agree with the reviewer, we have moved the Figure S18 into the manuscript as Figure 3.

13. “Some of the p values in these data figures should be checked. In particular, in SI Figure S18 the data for saccharide IVA appears to have a large enough standard deviation that the p value should be > 0.0029.”

In this paper, the significant differences were determined using an unpaired t test, and the p value was determined by two-tailed unpaired t test. $P < 0.05$ was considered statistically significant. We have rechecked the p value of all disaccharides of liver tissue, the p value of disaccharide IVA is 0.0029.

14. "Similarly, for SI Figure S19 "ns" should be clearly defined as a p value greater than a certain number, or the actual p values should just be provided for clarity."

Thanks for the reviewer's suggestion. We have defined "ns" as "not significant ($p > 0.05$)" in the Figure 3, Figure S18 and Figure S19 legend, respectively.

15. "The current description of results from the plasma study unsatisfactorily leaves several questions unanswered: what was the "range of concentrations" one could determine of the heparan sulfate hexasaccharide? What constitutes a "sufficiently high" concentration?"

We agree with those concerns. We can explain as follows.

In this paper, a series of concentration of HS hexasaccharide (6mer) from 1 $\mu\text{g/mL}$ to 32 $\mu\text{g/mL}$ were mixed with ^{13}C -labeled 6mer with concentration in 4 $\mu\text{g/mL}$ to evaluate the reliability and robustness of determining the plasma concentration of 6mer with ^{13}C -labeled 6-mer as internal standard. We will add the detailed information about the range of concentration of 6mer in the revised manuscript.

As a rare modification present in HS, the availability of 3-O-sulfated oligosaccharides is very limited. Therefore, the detection and quantification of 3-O-sulfated oligosaccharides from biological sources is challenged and difficult. In our paper, for the 3-O-sulfated 6mer, we can quantify the concentration as low as 1 $\mu\text{g/mL}$ in the plasma.

16. "In the linear dynamic range curves, which establish a linear dependence of peak area on disaccharide concentration, why do some of the data points have error bars and others do not? Were only certain concentrations tested multiple times? While the linear fits are convincing the specifics of data acquisition should be more clearly discussed."

In the linear dynamic range determination experiments, a series concentration of the mixture of the eight native disaccharides (1, 5, 10, 50, 100, 200, 400 and 800 $\mu\text{g/mL}$) were mixed with 10 $\mu\text{g/mL}$ eight ^{13}C -labeled DCs. In our paper, to obtain robust and reliable calibration curve, three replicates of each concentration were performed on LC-MS/MS analysis, and the curve and linear equation was determined based on ^{13}C -labeled DC normalized peak area ratio as a function of concentration for each native disaccharide. The error bars represent the standard deviation (S.D.) of the data. Some data have small error bars, so they are almost invisible in the figure.

We add those details about how to obtain the calibration curves under “Supplementary Information” in the revised manuscript.

17. “The seven isotopically labeled sulfo-disaccharides exhibit highly variable retention times, ranging from 17-47 mins despite similar elution conditions—this should be explained since their isolation and purity is a key requirement for the manuscript.”

We agree with this suggestion and provide more detailed explanation about elution of eight ¹³C-labeled disaccharides on the strong anion exchange column (SAX) Pro Pac PA1 (9x250 mm, Thermo Fisher Scientific).

As SAX-HPLC separation is based on charge density of analytes, the retention times of eight ¹³C-labeled disaccharides increase with the increase of sulfo groups, the less sulfated disaccharide $\Delta[^{13}\text{C}]\text{UA-GlcNAc}$ has the shortest retention time, while disaccharide $\Delta[^{13}\text{C}]\text{UA2S-GlcNS6S}$ has longest retention time. The different retention times of two groups isomers ($\Delta[^{13}\text{C}]\text{UA2S-GlcNS}$ and $\Delta[^{13}\text{C}]\text{UA-GlcNS6S}$; $\Delta[^{13}\text{C}]\text{UA2S-GlcNAc}$ and $\Delta[^{13}\text{C}]\text{UA-GlcNAc6S}$) on SAX-HPLC were determined by comparing with the retention times of the native disaccharide standards (Iduron) on SAX column.

18. “Several typographical errors were found, which should be corrected in the final version of the manuscript:

- o Page 4: “most of the disaccharides” insert “the”
- o Page 6: “leads to resistance to” change “resistant” to “resistance”
- o Page 7: “Large-scale synthesis of DC...” delete “For” at the start of sentence
- o Page 8: “role towards understanding the relationship” move “the”

We have changed those errors in the revised manuscript.

Reference

1. Copeland, R. J.; Balasubramaniam, A.; Tiwari, V.; Zhang, F.; Bridges, A.; Linhardt, R. J.; Shukla, D.; Liu, J., Using a 3-O-sulfated heparin octasaccharide to inhibit the entry of herpes simplex virus 1 *Biochemistry* **2008**, *47*, 5774-5783.